# Structures of a mammalian TRPM8 in closed state

Cheng Zhao[1,11], Yuan Xie [2,11✉], Lizhen Xu[3,11], Fan Ye [1], Ximing Xu[4], Wei Yang[1,5], Fan Yang [3,5,6,7✉] & Jiangtao Guo [1,5,7,8,9,10✉]

Transient receptor potential melastatin 8 (TRPM8) channel is a $Ca^{2+}$-permeable non-selective cation channel that acts as the primary cold sensor in humans. TRPM8 is also activated by ligands such as menthol, icilin, and phosphatidylinositol 4,5-bisphosphate ($PIP_2$), and desensitized by $Ca^{2+}$. Here we have determined electron cryo-microscopy structures of mouse TRPM8 in the absence of ligand, and in the presence of $Ca^{2+}$ and icilin at 2.5–3.2 Å resolution. The ligand-free state TRPM8 structure represents the full-length structure of mammalian TRPM8 channels with a canonical S4-S5 linker and the clearly resolved selectivity filter and outer pore loop. TRPM8 has a short but wide selectivity filter which may account for its permeability to hydrated $Ca^{2+}$. $Ca^{2+}$ and icilin bind in the cytosolic-facing cavity of the voltage-sensing-like domain of TRPM8 but induce little conformational change. All the ligand-bound TRPM8 structures adopt the same closed conformation as the ligand-free structure. This study reveals the overall architecture of mouse TRPM8 and the structural basis for its ligand recognition.

[1] Department of Biophysics, and Department of Neurology of the Fourth Affiliated Hospital, Zhejiang University School of Medicine, Hangzhou, Zhejiang 310058, China. [2] Department of Neurosurgery, Xijing Hospital, Fourth Military Medical University, Xi'an, Shaanxi 710032, China. [3] Department of Biophysics and Kidney Disease Center, The First Affiliated Hospital, Zhejiang University School of Medicine, Hangzhou, Zhejiang 310058, China. [4] Key Laboratory of Marine Drugs of Ministry of Education, School of Medicine and Pharmacy, Ocean University of China, Qingdao, Shandong 266003, China. [5] NHC and CAMS Key Laboratory of Medical Neurobiology, MOE Frontier Science Center for Brain Science and Brain-machine Integration, School of Brain Science and Brain Medicine, Zhejiang University, Hangzhou, China. [6] Alibaba-Zhejiang University Joint Research Center of Future Digital Healthcare, Hangzhou, Zhejiang 310058, China. [7] Liangzhu Laboratory, Zhejiang University Medical Center, 1369 West Wenyi Road, Hangzhou, Zhejiang 311121, China. [8] State Key Laboratory of Plant Physiology and Biochemistry, College of Life Sciences, Zhejiang University, Hangzhou, Zhejiang 310058, China. [9] Department of Cardiology, Key Laboratory of Cardiovascular Intervention and Regenerative Medicine of Zhejiang Province, Sir Run Run Shaw Hospital, Zhejiang University School of Medicine, Hangzhou, Zhejiang 310016, China. [10] Cancer Center, Zhejiang University, Hangzhou, Zhejiang 310058, China. [11]These authors contributed equally: Cheng Zhao, Yuan Xie, Lizhen Xu. ✉email: xieyuan2006@zju.edu.cn; fanyanga@zju.edu.cn; jiangtaoguo@zju.edu.cn

Transient receptor potential melastatin 8 (TRPM8) channel, a member of the TRP channel superfamily, is a cold-activated nonselective cation channel that acts as a primary cold receptor in humans in response to moderately cool temperatures (15–28 °C)[1–7]. In addition, TRPM8 is regulated by intracellular signal molecules. For example, intracellular $Ca^{2+}$ desensitizes TRPM8 (ref. [1]) and phosphatidylinositol 4,5-bisphosphate (PIP$_2$) potentiates the activation of TRPM8 (ref. [8]). TRPM8 can also be activated by both natural and synthetic cooling compounds such as menthol and icilin[6,7,9]. TRPM8 has been reported to play essential roles in oxaliplatin or nerve injury-induced cold allodynia, migraine, as well as inflammation-caused cold hypersensitivity[10–13]. Therefore, revealing the structural basis of the ligand modulation of TRPM8 will contribute to rational drug design against cold-related pain[14].

Recently, structures of TRPM8 from two birds, namely *Ficedula albicollis* (FaTRPM8) and *Parus major* (PmTRPM8), which both share 82% sequence identity and 91% sequence similarity to mouse TRPM8 (MmTRPM8) (Supplementary Fig. 1), have been determined in both ligand-free and agonist- or antagonist-bound states[15–17]. These studies provide a glimpse of the overall architectures of TRPM8. FaTRPM8 and PmTRPM8 have two features that differ from those of other mammalian TRPM channels such as TRPM4, TRPM5, and TRPM7 whose structures have been reported[18–22]. First, in the ligand-free states of bird TRPM8 structures (FaTRPM8$_{ligand-free}$, PDB: 6BPQ; PmTRPM8$_{ligand-free}$, PBD: 6O6A), the linker helix (S4-S5 linker) that connects transmembrane helices S4 and S5 forms a single straight helix with S5; in addition, the pore helices are resolved in low resolution, and selectivity filters and outer pore loops are invisible, likely due to their dynamics (Supplementary Fig. 2a, b). Second, in the $Ca^{2+}$-bound (PmTRPM8$_{Ca}$, PDB: 6O77) or $Ca^{2+}$-icilin-phosphatidylinositol 4,5-bisphosphate (PIP$_2$)-bound state (FaTRPM8$_{Ca-icilin-PIP2}$, PDB: 6NR3), the S4-S5 linker and S5 helices are restored to the canonical conformation, along with concerted structural rearrangement involving all transmembrane helices, including voltage-sensing-like domain (VSLD), S5, pore helix, S6, and TRP helix, and in PmTRPM8$_{Ca}$ with the reconstruction of filter and outer pore loop (Supplementary Fig. 2c–f). Based on these structures of bird TRPM8 in ligand-free and ligand-bound states, molecular mechanisms of icilin-PIP$_2$-induced activation and $Ca^{2+}$-induced desensitization of TRPM8 were proposed[16,17]. Since these bird TRPM8 structures display different features from those of other mammalian TRPM channels, whether these structures and the proposed ligand modulation mechanisms are conserved in the TRPM8 family across vertebrates including mammals needs further validation.

To reveal ligand modulation mechanisms of TRPM8, here we have performed a systematic structural analysis of mouse TRPM8 (MmTRPM8) and determined cryo-EM structures of MmTRPM8 in the absence of ligand, and in the presence of $Ca^{2+}$ and icilin.

## Results

**Structure determination of MmTRPM8 in detergent and lipid nanodisc.** We first determined MmTRPM8 structures in the detergent Lauryl Maltose Neopentyl Glycol (LMNG) in the absence of ligand at 3.0 Å resolution (MmTRPM8$_{LMNG-ligand-free}$, Supplementary Fig. 3), and in the presence of $Ca^{2+}$ at 2.9 Å resolution (MmTRPM8$_{LMNG-Ca}$, Supplementary Fig. 4), $Ca^{2+}$ + icilin at 3.0 Å resolution (MmTRPM8$_{LMNG-Ca-icilin}$, Supplementary Fig. 5), and $Ca^{2+}$ + icilin + PIP$_2$ at 3.2 Å resolution (MmTRPM8$_{LMNG-Ca-icilin-PIP2}$, Supplementary Fig. 6). The ligand-free MmTRPM8 sample was obtained by adding 2 mM EGTA to chelate the trace amount of free $Ca^{2+}$. To mimic the lipid environment of TRPM8 in the membrane, we prepared the MmTRPM8 nanodisc sample by reconstituting MmTRPM8 with MSP1 and the mixed lipid (POPC: POPG: POPE = 3: 1: 1, molar ratio) at a final molar ratio of 1: 2.5: 15. For the nanodisc sample with PIP$_2$, PIP$_2$ was first added to the mixed lipid with a mass ratio of 1: 1, and the molar ratio of MmTRPM8: MSP1: lipid was changed to 1: 2.5: 30. We then determined the 2.5 Å-resolution structure of MmTRPM8 in nanodisc in the presence of $Ca^{2+}$ + icilin (MmTRPM8$_{nanodisc-Ca-icilin}$, Supplementary Fig. 7) and the 3.0 Å-resolution structure of MmTRPM8 in nanodisc in the presence of $Ca^{2+}$ + icilin + PIP$_2$ (MmTRPM8$_{nanodisc-Ca-icilin-PIP2}$, Supplementary Fig. 8).

For all six MmTRPM8 structures, the cryo-EM density maps are of high quality, particularly in the transmembrane region, allowing us to build the model of major parts of MmTRPM8 (Fig. 1a, b). The S1–S6 transmembrane helices, pore helix,

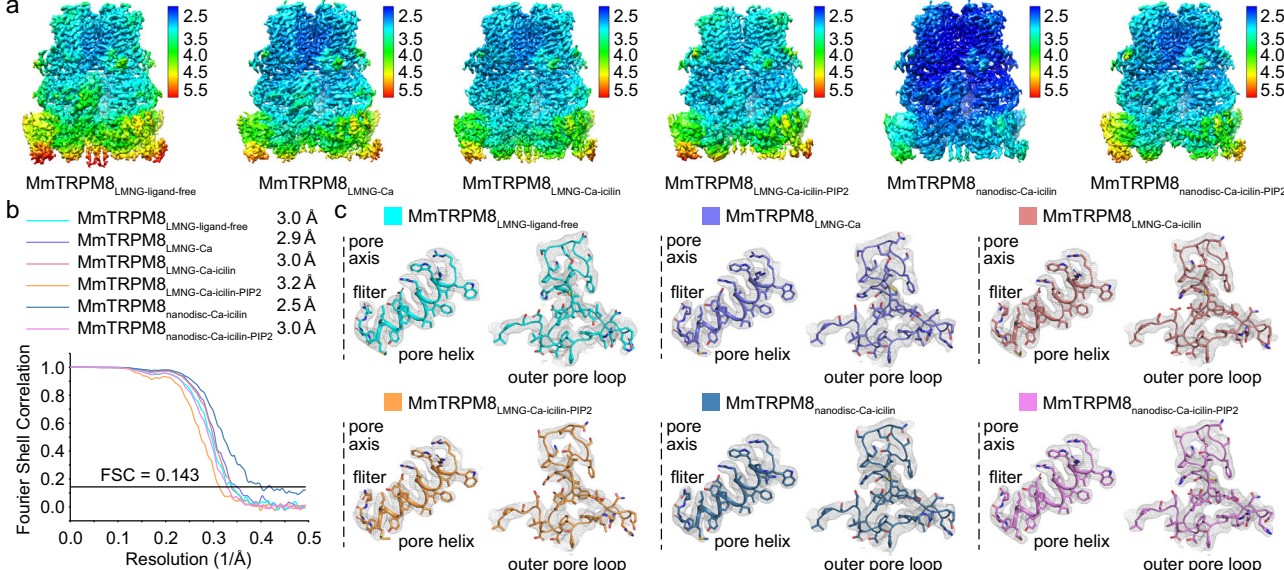

**Fig. 1 Structure determinations of MmTRPM8. a** The 3D reconstructions of MmTRPM8 colored by local resolutions in Å. **b** The Gold standard Fourier Shell Correlation (FSC) curves of the final 3D reconstructions of MmTRPM8 in different ligand-bound states. Source data are provided as a Source Data file. **c** Local maps of MmTRPM8 structures in the regions of pore helix, selectivity filter, and outer pore loop at the contour level of 3.5 σ.

**Table 1 Major information of six MmTRPM8 structures reported in this study.**

| Structures | TRPM8$_{LMNG-ligand-free}$ | TRPM8$_{LMNG-Ca}$ | TRPM8$_{LMNG-Ca-icilin}$ | TRPM8$_{LMNG-Ca-icilin-PIP2}$ | TRPM8$_{nanodisc-Ca-icilin}$ | TRPM8$_{nanodisc-Ca-icilin-PIP2}$ |
|---|---|---|---|---|---|---|
| Resolution (Å) | 3.0 | 2.9 | 3.0 | 3.2 | 2.5 | 3.1 |
| Ligands in the sample | None | $Ca^{2+}$ | $Ca^{2+}$, icilin | $Ca^{2+}$, icilin, PIP$_2$ | $Ca^{2+}$, icilin | $Ca^{2+}$, icilin, PIP$_2$ |
| Ligands modeled in the structure | None | $Ca^{2+}$ | $Ca^{2+}$, icilin | $Ca^{2+}$, icilin | $Ca^{2+}$, icilin | $Ca^{2+}$, icilin |
| Gate conformation | Closed | Closed | Closed | Closed | Closed | Closed |

selectivity filter, outer pore loop, and TRP helix are all well resolved (Fig. 1c). The ligands $Ca^{2+}$ and icilin are clearly assigned in the corresponding structures of TRPM8, whereas no PIP$_2$ molecule is unambiguously identified in the structures of either MmTRPM8$_{LMNG-Ca-icilin-PIP2}$ or MmTRPM8$_{nanodisc-Ca-icilin-PIP2}$, as discussed below. Table 1 summarized major information of the six MmTRPM8 structures (Table 1; Supplementary Table 1). In the following analyses, unless otherwise mentioned, we will focus on the ligand-free structure MmTRPM8$_{LMNG-ligand-free}$.

**The overall structure**. MmTRPM8 is a homotetramer with dimensions of $140 \times 110 \times 110$ Å. Four subunits assemble into a functional channel, with all the N-terminal region, the transmembrane domain, and the C-terminal region involved in the tetrameric assembly (Fig. 2). Each MmTRPM8 subunit contains multiple domains, including the N-terminal four TRPM homology repeats (MHR1 to MHR4), S1–S6 six transmembrane helices in a domain swap configuration, TRP helix, and C-terminal coiled-coil (Fig. 2c, d). This MmTRPM8 structure largely resembles structures of other mammalian TRPM channels such as TRPM4, TRPM5, and TRPM7 (refs. [18–22]).

**The ion conduction pore**. The whole ion conduction pore of MmTRPM8, consisting of S5, pore helix, selectivity filter, outer pore loop, and S6, is resolved at the highest resolution (Fig. 1a). The long outer pore loop (Val915–Pro952) between the filter and S6 forms an extracellular turret and encircles a deep vestibule at the external entrance of the channel (Fig. 3a, b). Multiple negatively charged residues reside on the inner surface of the vestibule thus favoring the recruitment of cations to the pore (Fig. 3a, b). The architecture of the outer pore loop is stabilized by extensive interactions with the S5 C-terminal end, pore helix, and S6 N-terminal end, as well as one disulfide bond within the outer pore loop (Fig. 3a).

In MmTRPM8, the selectivity filter is lined by residues $^{912}$FGQ$^{914}$ (Fig. 3c). The backbone carbonyls of Phe912 and Gly913 along with the Gln914 side chain, build the ion conduction pathway with a minimum atom-to-atom diameter of ~9 Å between both Phe912 and Gly913 carbonyl oxygen atoms, suggesting the passage of hydrated ions during ion conduction (Fig. 3c). In the maps of MmTRPM8$_{LMNG-ligand-free}$ and MmTRPM8$_{nanodisc-Ca-icilin-PIP2}$, but not others, sphere-shaped density peaks were clearly visible in the center of the selectivity and tentatively modeled as $Na^+$ ions, the cation at the highest concentration in the sample (Fig. 3d). Distances between the putative $Na^+$ and Gly913 carbonyl oxygens are ~4.5 Å, too far for direct ion coordination, supporting the passage of hydrated ions in the filter.

MmTRPM8 shares high sequence and structure similarity with mouse TRPM4 (MmTRPM4) at the filter (Fig. 3e)[18]. However, TRPM8 is permeable to $Ca^{2+}$ with the $P_{Ca}/P_{Na}$ of ~3 (ref. [5]), while TRPM4 is a monovalent-selective channel and is impermeable to $Ca^{2+}$ (refs. [18,23]). In the MmTRPM4 filter, the Gln973 side chain forms a strong hydrogen bond with the

carbonyl of Gly972 from its neighboring subunit, stabilizing the filter configuration with a minimum atom-to-atom diameter of 7.5–8 Å (Fig. 3e). In comparison, in the MmTRPM8 filter, the Gln914 side chain does not form a similar hydrogen bond with Gly913 carbonyl from its neighboring subunit and the atom-to-atom diameter is 1 Å larger than that in MmTRPM4 (Fig. 3c). As $Ca^{2+}$ has a larger hydration radius than $K^+$ and $Na^+$, the different filter diameters between MmTRPM8$_{LMNG-ligand-free}$ and MmTRPM4 may partially account for their difference in ion selectivity.

**The voltage-sensing-like domain (VSLD) and TRP helix**. The transmembrane helices S1–S4 comprise the VSLD, which maintains a cytosolic-facing hydrophilic cavity and provides potential ligand-binding sites[15]. VSLD interacts with the pore domain in three ways. First, S4 forms extensive hydrophobic packing with S5 from the adjacent subunit (Fig. 4a). Second, VSLD is covalently connected to the pore domain by the S4-S5 linker, which forms a ~150° smooth turn with S5 at the residue Met863 (Fig. 4b). Third, the TRP helix, which extends from the C-terminal end of S6 with a ~120° turn at the residue Val986 and runs underneath the S4-S5 linker towards the cytosolic-facing cavity in VSLD, also links VSLD and the pore domain (Fig. 4c). The TRP helix is tightly coupled to the S4–S5 linker through hydrophobic packing. Two polar residues in the C-terminal end of the TRP helix, namely Glu1004 and Try1005, point to this cytosolic-facing cavity of VSLD with their side chains. In addition, S4 adopts a 3$_{10}$ helical conformation at residues $^{841}$LRL$^{843}$, among which Arg842 provides positive charges for the cytosolic-facing cavity (Fig. 4b). Thus, ligands binding at this cytosolic-facing cavity may induce a local conformational change of C-terminal ends of S4 and (or) TRP helix, which is then transferred to the pore domain and regulates the gating of MmTRPM8 via S4-S5 linker and the TRP helix.

**Structural basis for the $Ca^{2+}$ recognition**. In the map of MmTRPM8$_{LMNG-Ca}$, strong density shows that a $Ca^{2+}$ ion is harbored by the cytosolic-facing cavity in the VSLD (Fig. 5a). In comparison, no density is observed at an equivalent site in the structure of MmTRPM8$_{LMNG-ligand-free}$ (Fig. 5b). The $Ca^{2+}$ is coordinated by side chains of residues Glu782, Gln785, Asn799, and Asp802 from S2 and S3 (Fig. 5a). In addition, the side chain of Tyr793 from the S2-S3 linker is positioned in proximity (Fig. 5a). The essential role of these $Ca^{2+}$ coordinating residues in $Ca^{2+}$-induced desensitization has been confirmed in PmTRPM8 (ref. [16]). This $Ca^{2+}$ binding site, initially revealed in human TRPM4 structure[20], is conserved in a subgroup of TRPM channels consisting of TRPM2 (ref. [24]), TRPM4 (ref. [20]), TRPM5 (ref. [22]), and TRPM8 (ref. [16,17]).

To confirm this $Ca^{2+}$ binding site in MmTRPM8, we performed all-atom molecular dynamic (MD) simulations using the $Ca^{2+}$-bound structure MmTRPM8$_{LMNG-Ca}$ as an initial model. Throughout 250 ns simulations, the $Ca^{2+}$ binds stably at this site, with a Root Mean Square Deviation (RMSD) within 3 Å

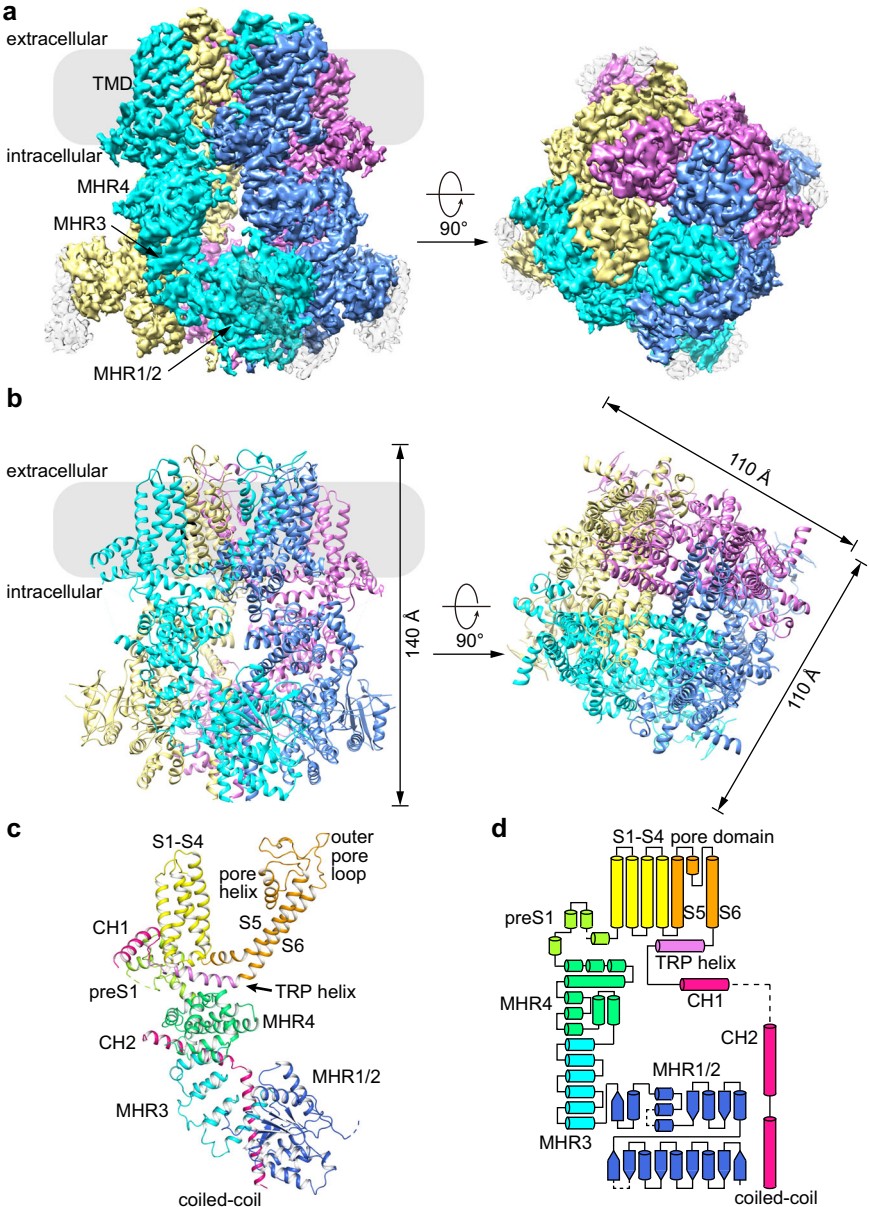

**Fig. 2 Overall structure of MmTRPM8_LMNG-ligand-free. a** The 3D reconstruction of MmTRPM8_LMNG-ligand-free with each subunit individually colored. **b** The cartoon representation of MmTRPM8_LMNG-ligand-free with each subunit individually colored. **c** Cartoon representation of one subunit of MmTRPM8_LMNG-ligand-free with domains colored by the rainbow. **d** The topology of one MmTRPM8 subunit.

(Fig. 5c). The distances between $Ca^{2+}$ and CG atoms of its coordinating residues Glu782, Asn799, and Asp802 remain 2–5 Å in most time of the simulations (Fig. 5d–f). Thus, the simulation results support the binding of one $Ca^{2+}$ ion in this cytosolic-facing cavity in the VSLD of MmTRPM8.

$Ca^{2+}$ potentiates the icilin activation on TRPM8 (ref. [9]). To functionally validate this $Ca^{2+}$ binding site, we tested the $Ca^{2+}$ effect on the icilin-induced currents of WT and $Ca^{2+}$-coordinating residue mutants of MmTRPM8 using a whole-cell patch-clamp recording. Among the 19 mutants we tested, five maintained the icilin sensitivity (Fig. 5g). While the $Ca^{2+}$ amplified the icilin-evoked current of WT MmTRPM8 by 2-fold, it did not increase the currents of the five mutants, namely Q785K, Q785Y, N799R, N799L, and D802K (Fig. 5h, Supplementary Fig. 9a), likely due to the loss of $Ca^{2+}$-interacting residues. These electrophysiological data further confirm the assignment of $Ca^{2+}$ in the VSLD of MmTRPM8.

**Structural basis for the icilin recognition.** In the structure of MmTRPM8_LMNG-Ca-icilin, well-defined bulky density in the cytosolic-facing cavity of VSLD allows us to confidently model the icilin molecule (Fig. 6a, b). The orientation of icilin is assigned based on the shape of the nitro group, which forms interaction with the Phe839 side chain (Fig. 6b, c). Icilin is also stabilized by anion-π interaction between the Asp802 side chain and the aromatic ring of the nitrophenyl moiety, as well as the hydrogen bond between the Arg842 side chain and the hydroxyl group in the hydroxyphenyl moiety (Fig. 6c).

In the structure of MmTRPM8_LMNG-Ca-icilin, the $Ca^{2+}$ resides in the vicinity but does not directly interact with icilin. Instead, its coordinating residue Asp802 stabilizes the icilin by anion-π interaction. In addition, Asp802 helps fix the orientation of Arg842 side chain by forming salt bridges (Fig. 6c). Upon icilin binding, the Arg842 side chain goes down towards the hydroxyphenyl moiety, making space for the nitrophenyl and

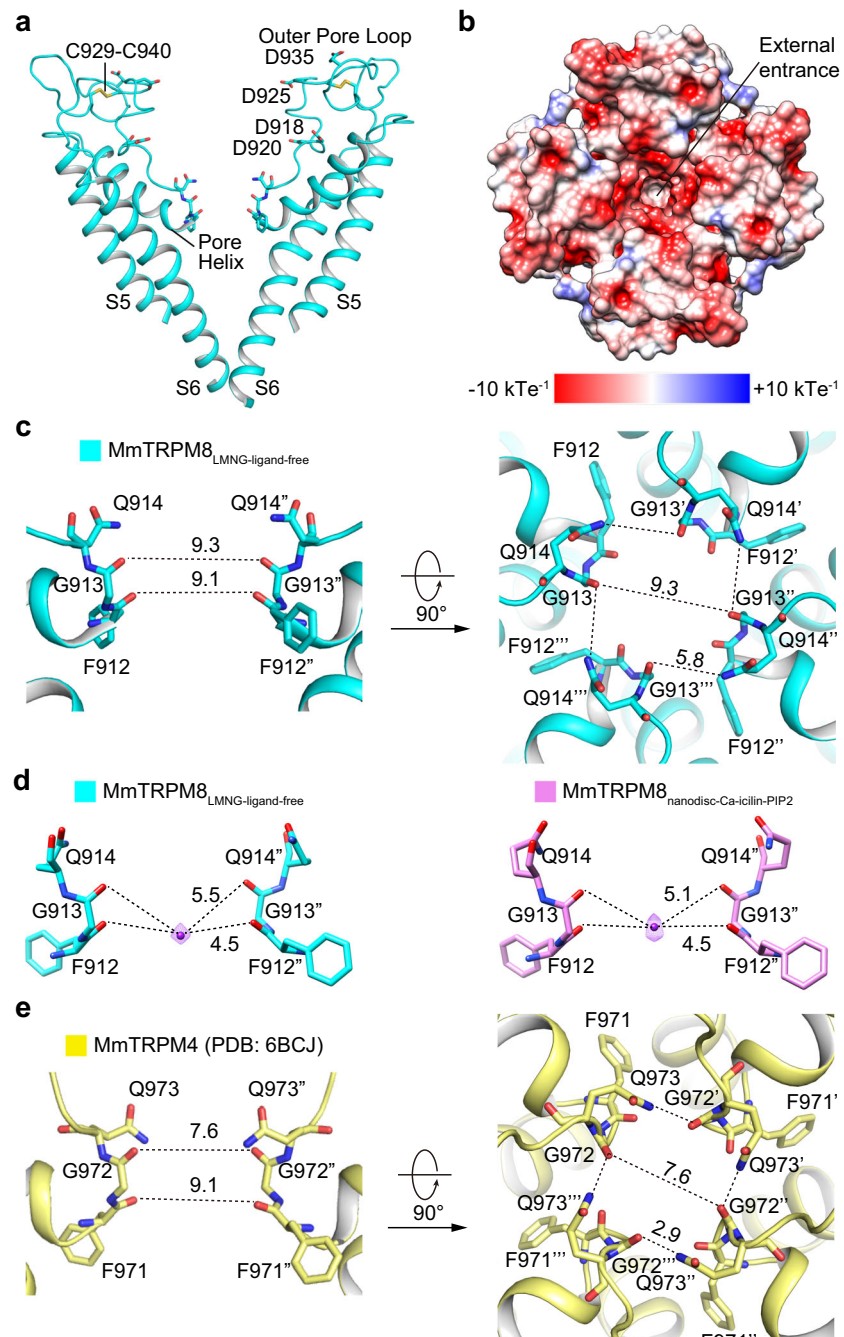

**Fig. 3 Ion conduction pore of MmTRPM8_LMNG-ligand-free. a** Ion conduction pore of MmTRPM8_LMNG-ligand-free with front and rear subunits removed for clarity. **b** Top view of the surface electrostatic potential at the external entrance of MmTRPM8_LMNG-ligand-free. **c** The selectivity filter of MmTRPM8_LMNG-ligand-free. Dash lines show atom-to-atom cross distances in Å. The 4.7 Å distance shows that the Gln914 side chain does not form a hydrogen bond with Gly913 from its neighboring subunit. **d** Sphere-shaped densities in the selectivity filters of MmTRPM8_LMNG-ligand-free and MmTRPM8_nanodisc-Ca-icilin-PIP2. **e** The selectivity filter of MmTRPM4 (PDB: 6BCJ). The Gln973 side chain forms a strong hydrogen bond with the carbonyl of Gly972 from its neighboring subunit.

pyrimidine moieties, and meanwhile forms a hydrogen bond with the hydroxyl group in the hydroxyphenyl moiety (Fig. 6c, d). Therefore, $Ca^{2+}$ potentiates the icilin activation through the Asp802 and Asp802-Arg842 interaction networks.

To confirm the pose of icilin modeled in the structure of MmTRPM8_LMNG-Ca-icilin, we performed all-atom MD simulations using the structure of MmTRPM8_LMNG-Ca-icilin as an initial model. Throughout the 250 ns simulation, the icilin binds stably at its original site, with an RMSD within 4 Å (Fig. 6e), and a

distance of 3–7 Å between the N21 atom of icilin and the CG atom of Phe839 (Fig. 6f). Meanwhile, the $Ca^{2+}$ also binds stably at its original site, with an RMSD of within 3 Å (Fig. 6g), and a distance of 3–4 Å between the $Ca^{2+}$ and the CG atom of Asp802 in most time of the simulations (Fig. 6h). These computational data provide additional support for our assignment of icilin based on the map density.

To validate the icilin binding pocket in MmTRPM8, we performed a whole-cell patch-clamp recording. Icilin activated

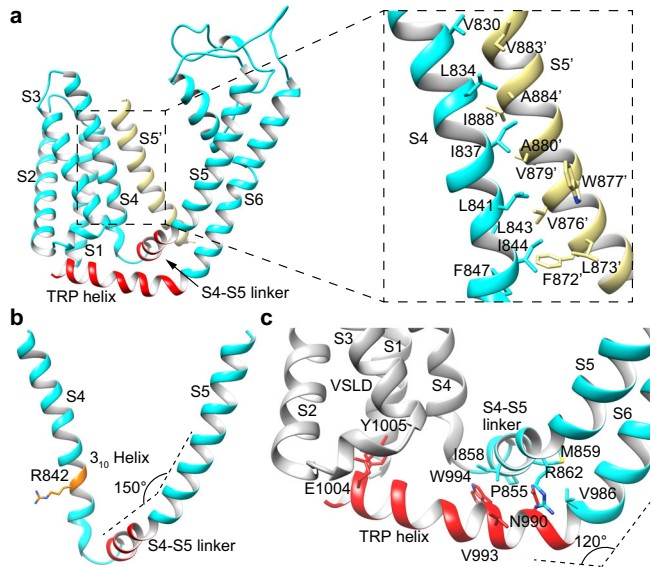

**Fig. 4 Interactions between VSLD and the pore domain in MmTRPM8_LMNG-ligand-free.** **a** Hydrophobic interactions between S4 (cyan) and S5 from the adjacent subunit (yellow). **b** The structure of S4 and S4-S5 linker. The $3_{10}$ helix in S4 is highlighted by orange color. **c** Interactions between TRP helix and S4-S5 linker. The polar residues Glu1004 and Tyr1005 in the C-terminal end of TRP helix point to the cytosolic-facing cavity of VSLD.

wild type (WT) MmTRPM8 with an $EC_{50}$ value of $401.0 \pm 43.9$ nM (Fig. 6i; Supplementary Fig. 9b; Supplementary Table 2), similar to the previous reports[5]. Introducing point mutations at residues Asp802, Phe839, and Arg842 within the icilin binding pocket largely perturbed the icilin activation, resulting in an increased $EC_{50}$, reduced current amplitude, or loss of activation effect (Fig. 6i–k). As a control, these MmTRPM8 mutants were able to be activated by menthol, indicating that they were still functional (Supplementary Fig. 9b; Supplementary Table 2). These observations demonstrate that residues within the binding pocket revealed by our cryo-EM structure are critical for icilin activation.

**All mouse TRPM8 structures are in the same closed state.** To reveal the ligand-induced conformational change of MmTRPM8, we performed structural alignments of MmTRPM8_LMNG-ligand-free with all MmTRPM8 structures in ligand-bound states, which yields an RMSD of 0.47–0.75 Å over 930 Cα atoms within one subunit (Supplementary Fig. 10). Therefore, all the six MmTRPM8 structures are essentially in the same conformation. Moreover, the consistency of MmTRPM8 structures determined in detergent micelle and lipid nanodisc indicates that these MmTRPM8 structures adopt a stable conformation.

To reveal in what states MmTRPM8 structures are, we calculated the pore radii of these MmTRPM8 structures along the ion conduction pathway using the program HOLE[25]. The activation gates of all MmTRPM8 structures are closed by the hydrophobic residue Val976, which forms the only constriction along ion conduction pore with a Van Der Waals radius less than 1 Å (Fig. 7a, b). To further analyze the conformation of the activation gate in MmTRPM8 structures, we align the MmTRPM8_LMNG-ligand-free structure with mouse TRPM4 (MmTRPM4) and TRPM7 (MmTRPM7) whose pore domains have been unambiguously resolved, at S6 (refs. 18,21). MmTRPM8_LMNG-ligand-free is superimposed well with closed MmTRPM4 (PDB: 6BCJ) and closed MmTRPM7 (PDB: 5ZX5), suggesting they adopt similar

conformations at the activation gate (Fig. 7c, d). In MmTRPM4 and MmTRPM7, the constrictions are formed by two layers of residues, namely Ile1036 and Ser1040, and Ile1093 and Asn1097, respectively, whereas in MmTRPM8_LMNG-ligand-free, equivalent residues are Val976 and Gly980 (Fig. 7e–g). Because of the loss of side chain in Gly980, MmTRPM8_LMNG-ligand-free has only one layer of constriction. Therefore, we propose that these six MmTRPM8 structures are all in the same closed state, similar to the closed states of MmTRPM4 and MmTRPM7 (refs. 18,21).

## Discussion

In this report, we present the structures of a mammalian TRPM8 in ligand-free, $Ca^{2+}$-bound, and $Ca^{2+}$-icilin-bound states either in LMNG detergent or lipid nanodisc condition. All these six structures maintain clearly resolved S1–S6, pore helix, selectivity filter, outer pore loop, and TRP helix (Fig. 1c, Supplementary Figs. 3–8), and adopt the same closed conformation (Fig. 7). TRPM8 is desensitized by $Ca^{2+}$ and activated by $Ca^{2+}$-icilin-$PIP_2$[1,8]. Although the $Ca^{2+}$ and icilin are well defined in the map, no $PIP_2$ molecule is unambiguously resolved. Since we are unable to capture the desensitized state and the open state, how $Ca^{2+}$ desensitizes and $Ca^{2+}$-icilin-$PIP_2$ activates TRPM8 remains unknown. We suspect that the current closed state is a low-energy stable state, which is easy to be achieved but difficult to be crossed over in vitro. In addition, the low open probability of TRPM8 at 0 mV (~0.2 to 0.5) further reduces the possibility to obtain the open-state structure[3]. Similarly, ligand-free and $Ca^{2+}$-bound structures of human TRPM4 are also captured in the same closed state[20]. In the future, determination of the agonist-bound open-state structure of TRPM8 will elucidate its ligand activation mechanism.

Nevertheless, our high-resolution MmTRPM8 structures clarify four key facts of TRPM8 in structures and mechanisms.

First, structurally MmTRPM8 is similar to other mammalian TRPM channels such as TRPM4, TRPM5, and TRPM7 in the transmembrane domain. In the ligand-free state, they all maintain the canonical S4-S5 linker and ordered pore helix, selectivity filter, and outer pore loop. The previously reported structure features in bird TRPM8 structures in the ligand-free state, such as low-resolution pore helix, invisible selectivity filter and outer pore loop, and one single straight helix formed by S4-S5 linker and S5 (Supplementary Fig. 2a, b), are not observed in the MmTRPM8 structure in the ligand-free state. Whether these structure features are unique in bird TRPM8 awaits further study. Obviously, they are not universal in the TRPM8 family, at least in MmTRPM8. A strict comparison of MmTRPM8_LMNG-ligand-free with PmTRPM8_ligand-free reveals structure differences distributed throughout the transmembrane domain (Supplementary Fig. 11a–f). In fact, an ion channel with stable selectivity tends to maintain a relatively rigid architecture of the filter, as revealed by the structures of potassium channel MthK[26], sodium channel NavAb[27], and calcium channel Cav1.1 (ref. 28). The collapse of the selectivity filter in some channels such as potassium channel KcsA[29] and sodium channel NavRh[30] results in inactivation. Under this circumstance, our MmTRPM8_LMNG-ligand-free structure with a clearly resolved filter may represent a physiologically-relevant state.

Second, MmTRPM8 essentially does not undergo conformational change upon the binding of $Ca^{2+}$ and icilin under our structure determination conditions in vitro. Previously, structure comparisons of PmTRPM8_ligand-free and PmTRPM8_Ca attributed the conformational difference to the desensitization induced by the $Ca^{2+}$ binding[16]. However, structural alignments show that the $Ca^{2+}$-bound PmTRPM8_Ca, instead of PmTRPM8_ligand-free, displays a similar conformation as MmTRPM8_LMNG-ligand-free, with an RMSD of 2.24 Å over 930 Cα atoms within one

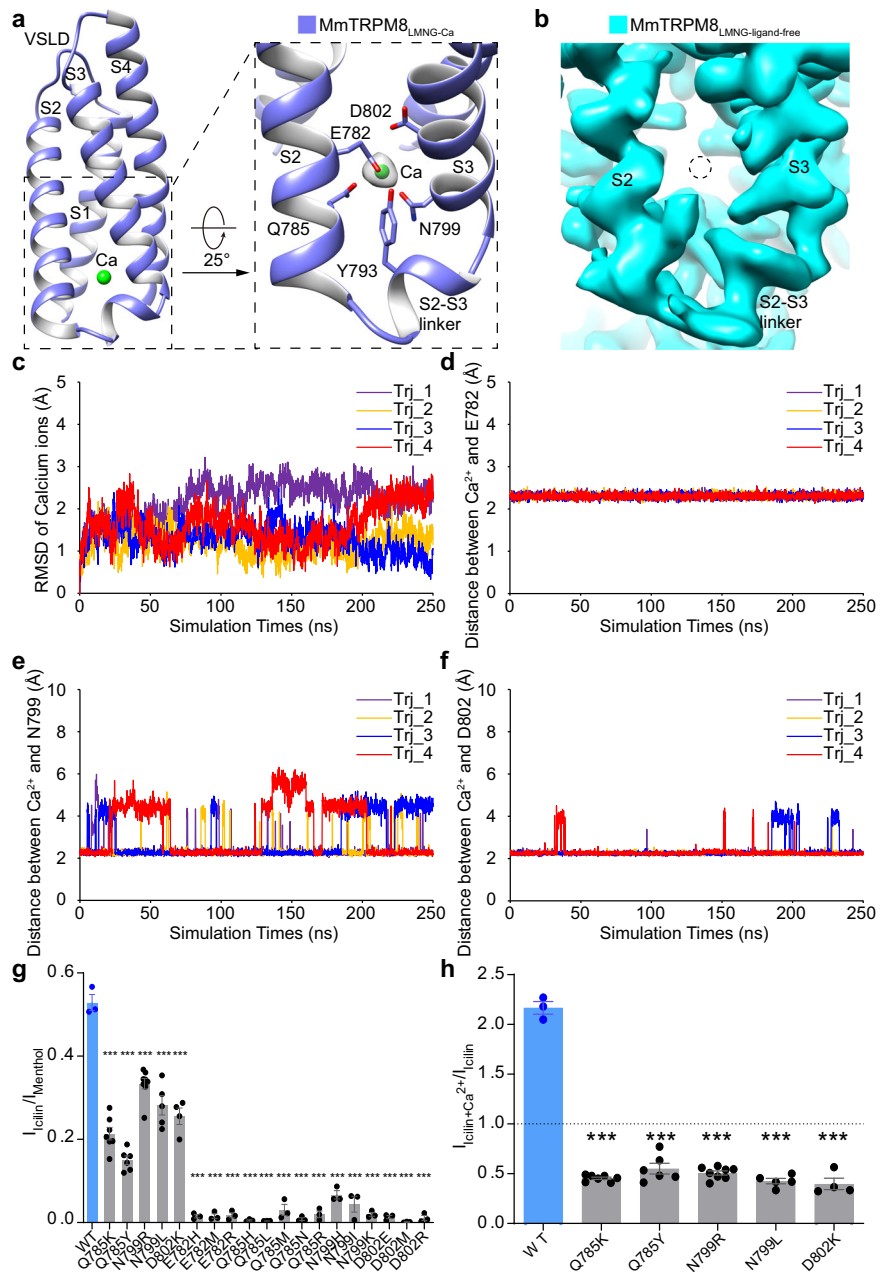

**Fig. 5 The Ca²⁺ binding site in MmTRPM8. a** The VSLD of MmTRPM8_LMNG-Ca harbors a Ca²⁺ binding site. The Ca²⁺ density is shown at the level of 0.013 in UCSF chimera. **b** The absence of density at the potential Ca²⁺ binding site in VSLD in the map of MmTRPM8_LMNG-ligand-free at the level of 0.013 in UCSF chimera. **c** The RMSD of the Ca²⁺ plotted against simulation time. The distances between the Ca²⁺ and CG atoms in Glu782 (**d**), Asn799 (**e**), and Asp802 (**f**) plotted against simulation time. **g** Summary of MmTRPM8 activated by menthol and icilin. In HEK293T cells expressing WT or mutant MmTRPM8, the application of menthol (2 mM) or icilin (10 μM) elicited outward currents. For the mutants proposed to be important for Ca²⁺ coordination, most of them were no longer activated by icilin. WT and mutants were colored in blue and grey, respectively. For WT MmTRPM8, $n = 3$; for Q785K, $n = 7$, $P = 2.25E{-}06$; for Q785Y, $n = 6$, $P = 2.41E{-}07$; for N799R, $n = 8$, $P = 1.78E{-}05$; for N799L, $n = 5$, $P = 3.46E{-}04$; for D802K, $n = 4$, $P = 1.99E{-}04$; for E782H, $n = 3$, $P = 1.23E{-}05$; for E782M, $n = 3$, $P = 1.34E{-}05$; for E782R, $n = 3$, $P = 1.41E{-}05$; for Q785H, $n = 3$, $P = 1.06E{-}05$; for Q785L, $n = 3$, $P = 1.04E{-}05$; for Q785M, $n = 3$, $P = 2.81E{-}05$; for Q785N, $n = 3$, $P = 1.14E{-}05$; for Q785R, $n = 3$, $P = 1.61E{-}05$; for N799H, $n = 3$, $P = 2.79E{-}05$; for N799I, $n = 3$, $P = 5.56E{-}05$; for N799K, $n = 3$, $P = 1.31E{-}05$; for D802E, $n = 3$, $P = 1.21E{-}05$; for D802M, $n = 3$, $P = 1.02E{-}05$; for D802R, $n = 3$, $P = 1.33E{-}05$. Two-sided $t$-test; *** denotes $P < 0.001$. All data points are mean ± s.e.m. **h** Summary of MmTRPM8 and mutants activated by icilin alone (10 μM) or in the presence of Ca²⁺ (1 mM). WT and mutants were colored in blue and grey, respectively. For WT MmTRPM8, $n = 3$; for Q785K, $n = 7$, $P = 1.62E{-}10$; for Q785Y, $n = 6$, $P = 3.87E{-}07$; for N799R, $n = 8$, $P = 1.02E{-}10$; for N799L, $n = 5$, $P = 1.03E{-}07$; for D802K, $n = 4$, $P = 4.98E{-}06$. Two-sided $t$-test; *** denotes $P < 0.001$). All data points are mean ± s.e.m. For **c–h**, source data are provided as a Source Data file.

subunit (Supplementary Fig. 11a, g). As analyzed above, all MmTRPM8 structures are in the same closed state, similar to the closed states of MmTRPM4 and MmTRPM7. Therefore, it is likely that PmTRPM8_Ca is also in the closed state (Supplementary

Fig. 11h), similar to the Ca²⁺-bound MmTRPM8_LMNG-Ca structure. Ca²⁺ binding at the VSLD in PmTRPM8 seems to stabilize the intact structure of the transmembrane domain. Without Ca²⁺, the selectivity filter and outer pore loop of PmTRPM8

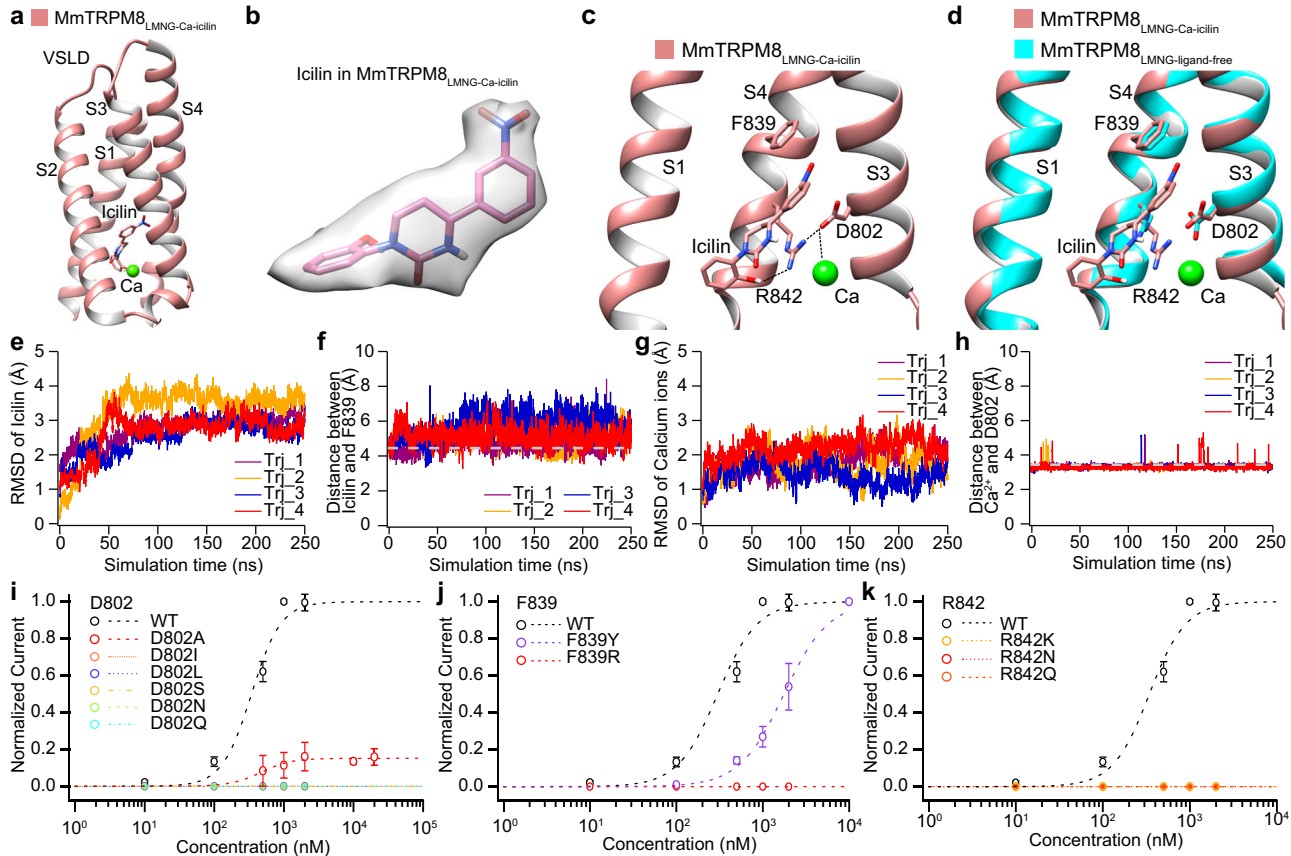

**Fig. 6 Structural basis and MD simulation for the icilin recognition. a** The Ca$^{2+}$ and icilin binding sites in the VSLD of MmTRPM8$_{LMNG-Ca-icilin}$. **b** The icilin density in MmTRPM8$_{LMNG-Ca-icilin}$ at the level of 0.015 in UCSF chimera. **c** Interactions between MmTRPM8 and Ca$^{2+}$ and icilin. **d** The binding of Ca$^{2+}$ and icilin induces local conformational change revealed by structural comparison of MmTRPM8$_{LMNG-Ca-icilin}$ (salmon) and MmTRPM8$_{LMNG-ligand-free}$ (cyan). **e** The RMSD of icilin plotted against simulation time. **f** The distance between the N21 atom in icilin and CG atom in Phe839 plotted against simulation time. **g** The RMSD of Ca$^{2+}$ plotted against simulation time. **h** The distance between the Ca$^{2+}$ and CG atom in Asp802 plotted against simulation time. **i–k** Concentration-dependent icilin activation of Asp802, Phe839, and Arg842 mutants in comparison with WT MmTRPM8 in whole-cell patch-clamp recordings (For WT MmTRPM8, $n = 5$; for D802A, D802I, D802L, D802S, D802N, D802Q, F839R, F839Y, R842K, R842N, and R842Q, $n = 3$). All data points are mean ± s.e.m. For **e–k**, source data are provided as a Source Data file.

become disordered, along with structural rearrangement of VSLD, S4-S5 linker, S6, and the TRP helix (Supplementary Fig. 2a, c, e).

Third, our higher-resolution structure of MmTRPM8$_{LMNG-Ca-icilin}$ confirms the binding configuration of icilin in MmTRPM8. In FaTRPM8$_{Ca-icilin-PIP2}$, the icilin was modeled at the same position, but in the opposite orientation, likely due to the resolution limit (Supplementary Fig. 12a, b)[17]. To further validate the orientation of icilin within its binding pocket in MmTRPM8, we mutated Tyr1005 to a Phe residue and measured icilin-induced current. If icilin binds to TRPM8 in the orientation as shown in the FaTRPM8$_{Ca-icilin-PIP2}$ structure (Supplementary Fig. 12c)[17], Tyr1005 would form a strong hydrogen bond with the nitro group of icilin, and the Y1005F mutation would disrupt this hydrogen bond and largely affect icilin activation. However, we observed that icilin robustly activated the Y1005F mutant of MmTRPM8 with a concentration-response curve virtually identical to that of the WT channel (Supplementary Fig. 12d, e; Supplementary Table 2). Therefore, we believe that icilin does not form a strong interaction with Tyr1005 as suggested by the structure of FaTRPM8$_{Ca-icilin-PIP2}$.

Fourth, we do not observe PIP$_2$ bound in the structure of either MmTRPM8$_{LMNG-Ca-icilin-PIP2}$ or MmTRPM8$_{nanodisc-Ca-icilin-PIP2}$. Like many other TRP channels, TRPM8 is activated by PIP$_2$. To capture an open-state structure, we added PIP$_2$ in the

MmTRPM8 sample in both detergent and nanodisc conditions, with the presence of Ca$^{2+}$ and icilin. Yet the identification of PIP$_2$ in the maps of both structures was unsuccessful. Previous structures of PIP$_2$-bound FaTRPM8 show that the PIP$_2$ binds at the cavity formed by the pre-S1 domain, S1, the junction of S4 and S5, and the TRP helix (Supplementary Fig. 13a). In the structures of MmTRPM8$_{LMNG-Ca-icilin-PIP2}$ or MmTRPM8$_{nanodisc-Ca-icilin-PIP2}$, bulk density corresponding to hydrophobic tails of lipid or detergent occupies the equivalent site of hydrophobic tails of PIP$_2$, but no density accounts for the inositol 1,4,5-trisphosphate head group (Supplementary Fig. 13b, c). The PIP$_2$ in MmTRPM8 seems very dynamic and is difficult to be observed under our structure determination conditions.

## Methods

**Protein expression and purification.** The full-length mouse TRPM8 cDNA was synthesized and cloned into a modified pEZT-BM vector in frame with a C-terminal GGSSGG linker followed by a strep tag II. Human Embryonic Kidney (HEK) 293 F suspension cells (Life Technologies) for heterologous TRPM8 expression were maintained at 37 °C in SMM 293-TI complete medium (Sino Biological Inc.) supplemented with 2% fetal bovine serum (FBS, Yeasen Biotechnology (Shanghai) Co., Ltd.). The P3 baculovirus was generated via the Bac-Mam system (Thermo Fisher Scientific) and used to transduce HEK293F cells at a cell density of $4 \times 10^6$ cells/mL. For induction, 10 mM sodium butyrate was added 12 h post-transduction, and cells were maintained at 30 °C to boost protein expression. Cells were harvested after 48 h, then flash-frozen in liquid nitrogen and stored at −80 °C.

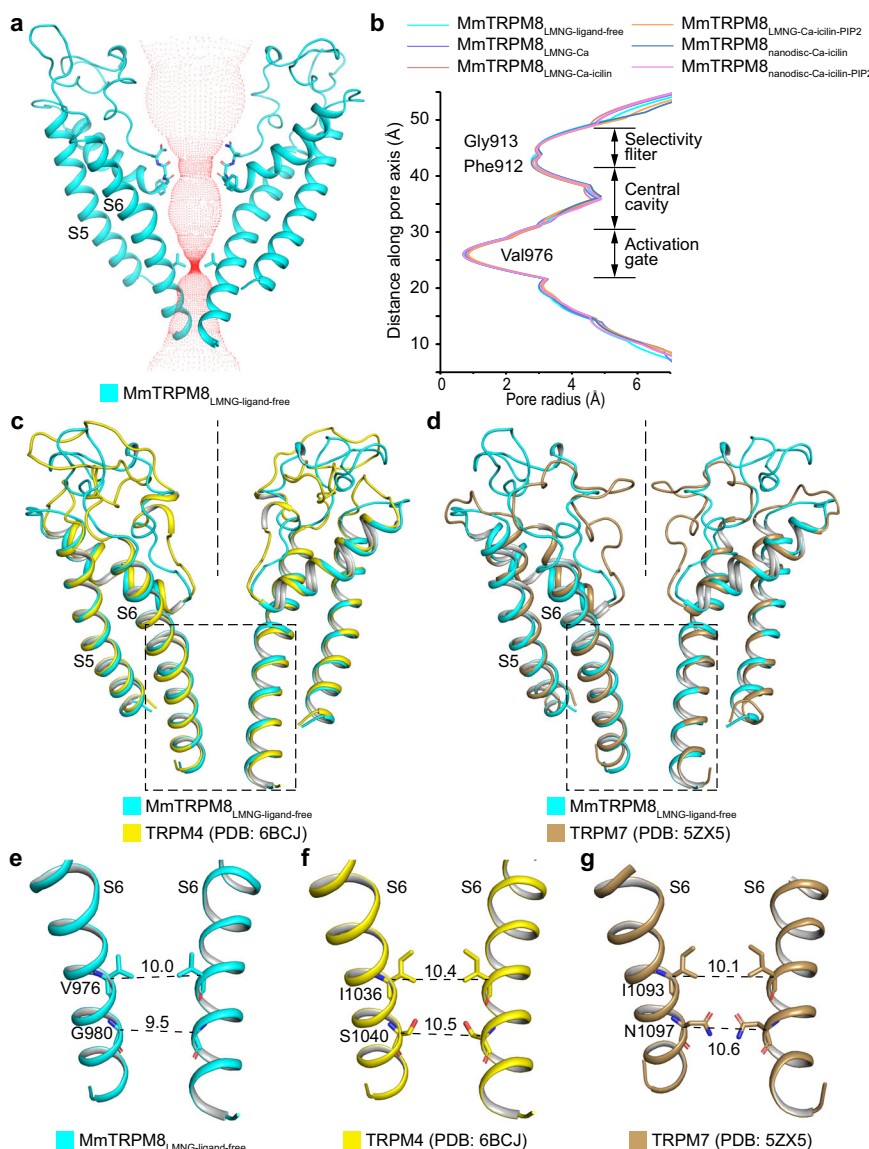

**Fig. 7 MmTRPM8 structures are in closed states. a** Ion conduction pore of MmTRPM8_LMNG-ligand-free with front and rear subunits removed for clarity. Central pathway is marked with a dotted mesh. **b** Pore radii along the central axis in six MmTRPM8 structures. Source data are provided as a Source Data file. **c** Structural comparison of the pore domain in MmTRPM8_LMNG-ligand-free and the closed MmTRPM4 (PDB: 6BCJ). For clarity, the front and rear subunits are omitted. The rectangular box indicates the activation gate. **d** Structural comparison of the pore domain in MmTRPM8_LMNG-ligand-free and the closed MmTRPM7 (PDB: 5ZX5). Structures of the activation gates in MmTRPM8_LMNG-ligand-free (**e**), closed MmTRPM4 (**f**), and closed MmTRPM7 (**g**). Dash lines show diagonal atom-to-atom distances of Cα atoms from constriction-lining residues (in Å).

A cell pellet from a 2 L culture was lysed by re-suspending in 40 mL buffer containing 20 mM Tris-HCl, pH 8.0, and a protease inhibitor cocktail (2 μg/mL DNase I, 0.5 μg/mL pepstatin, 2 μg/mL leupeptin, 1 μg/mL aprotinin, and 1 mM PMSF) for 1 h at room temperature. The lysate was supplemented with 150 mM NaCl and rotated for 20 min before being solubilized with 1% lauryl maltose neopentyl glycol (LMNG, Anatrace) and 0.2% cholesteryl hemisuccinate tris salt (CHS, Anatrace) for 2 h at room temperature. The insoluble cell fragment was removed by centrifugation at 48,000 $g$ for 50 min at 18 °C. The supernatant was incubated with 0.8 mL Strep-Tactin Sepharose resin (IBA) for 2 h at room temperature with gentle rotation. Beads were loaded onto a gravity column and washed with wash buffer containing 20 mM Tris-HCl, pH 8.0, 150 mM NaCl, 0.005% LMNG, and 0.001% CHS for 20 column volumes. The protein was then eluted with wash buffer containing 10 mM d-Desthiobiotin (Sigma) and further purified in a Superose 6 gel filtration column (GE Healthcare) in 20 mM Tris-HCl, pH 8.0, 150 mM NaCl, 0.0025% LMNG, and 0.0005% CHS. The peak fraction of TRPM8 was collected and concentrated in a 100-kDa concentrator (Amicon Ultra, Millipore Sigma) to about 4 mg/mL for cryo-EM sample preparation. For the ligand-free sample, 2 mM EGTA was added into the protein sample to chelate endogenous $Ca^{2+}$. For the $Ca^{2+}$-bound sample, no exogenous $Ca^{2+}$ was added and the bound $Ca^{2+}$ is from endogenous $Ca^{2+}$ or a trace amount of $Ca^{2+}$ in solution.

For the $Ca^{2+}$-icilin-bound sample, 200 μM icilin and 0.5 mM $CaCl_2$ were added and the mixture was incubated at room temperature for 30 min before grid preparation. For the TRPM8_LMNG-Ca-icilin-PIP2 sample, 1 mM $CaCl_2$ was added throughout the purification procedure. Protein was mixed with 200 μM icilin and 0.5 mM diC8-PIP2 (Avanti) and incubated at room temperature for 30 min before grid preparation.

For nanodisc sample preparation, mouse TRPM8 was fusion-expressed with an N-terminal maltose-binding protein (MBP) followed by a TEV protease cleavage sequence. Baculovirus preparation, protein expression, and affinity purification procedures were the same as previously mentioned. The protein eluted from Strep-Tactin Sepharose resin was concentrated and mixed with MSP1 and lipid (POPC: POPE: POPG = 3: 1: 1, molar ratio) at a molar ratio of 1: 2.5: 15. For the sample of MmTRPM8_nanodisc-Ca-icilin-PIP2, $PIP_2$ was pre-mixed with lipid at a mass ratio of 1: 1, and the molar ratio of MmTRPM8: MSP1: lipid was adjusted to 1: 2.5: 30. Detergents were removed by incubating the mixture of TRPM8, lipid, and MSP1 with Bio-Beads SM2 (Bio-Rad) at a concentration of 200 mg/mL by gentle agitation. Bio-Beads were replaced with fresh ones every 6 h for 2 times. TEV protease was added during the third time Bio-Beads incubation to remove MBP. After the removal of detergent, the protein was concentrated and injected into a Superose 6 gel filtration column (GE Healthcare) in 20 mM Tris-HCl, pH 8.0,

150 mM NaCl. Peak fraction was collected, concentrated, and mixed with 0.5 mM CaCl$_2$ and 200 μM icilin before grid preparation.

**Cryo-EM sample preparation and data acquisition.** For grids preparation, 3 μL TRPM8 protein was loaded onto glow-discharged R1.2/1.3 Quantifoil grids. Grids were blotted for 4.5 s at 4 °C under 100% humidity and plunge-frozen in liquid ethane using a Vitrobot Mark IV (FEI). Micrographs were acquired on a Titan Krios microscope (FEI) operated at a voltage of 300 kV with a K2 summit direct electron detector (Gatan) via SerialEM software following standard procedure. A calibrated magnification of 49,310 × was used for imaging, yielding a pixel size of 1.014 Å. Micrographs were dose-fractionated to 40 frames with a dose rate of 8 e − /pixel/s and a total exposure time of 8 s, corresponding to a total dose of ~62 e − /Å$^2$.

**Image processing.** The MotionCorr2 (ref. [31]) and the GCTF[32] programs were utilized for motion correction and CTF parameters estimation, respectively. All image processing steps were carried out with RELION 3.0 (ref. [33]).

For MmTRPM8$_{LMNG-ligand-free}$, 1,432 micrographs were collected and 324,027 particles were auto-picked and extracted with a binning factor of 3 before 2D classification. A total of 291,586 particles were selected for 2 rounds of 3D classification using the map of PmTRPM8$_{Ca}$ (PDB: 6O77) as the reference. Particles from five 3D classes were selected, combined, and re-extracted to the pixel size of 1.014 Å, followed by 3D refinement with C4 symmetry and particle polishing via RELION 3.0. The final 3D reconstruction of MmTRPM8$_{LMNG-ligand-free}$ from 40,653 particles yielded an EM map with a resolution of 2.98 Å.

For MmTRPM8$_{LMNG-Ca}$, 1884 micrographs were collected and 428,292 particles were auto-picked and extracted with a binning factor of 3 before 2D classification. A total of 403,154 particles were selected for 2 rounds of 3D classification using the MmTRPM8$_{LMNG-ligand-free}$ map as the reference. Particles from six 3D classes were selected, combined, and reextracted to the pixel size of 1.014 Å, followed by 3D refinement with C4 symmetry and particle polishing via RELION 3.0. The final 3D reconstruction of MmTRPM8$_{LMNG-Ca}$ from 53,900 particles yielded an EM map with a resolution of 2.88 Å.

For MmTRPM8$_{LMNG-Ca-icilin}$, 2079 micrographs were collected and 469,791 particles were auto-picked and extracted with a binning factor of 3 before 2D classification. A total of 435,438 particles were selected for 2 rounds of 3D classification using the MmTRPM8$_{LMNG-ligand-free}$ map as the reference. Particles from six 3D classes were selected, combined, and reextracted to the pixel size of 1.014 Å, followed by 3D refinement with C4 symmetry and particle polishing via RELION 3.0. The final 3D reconstruction of MmTRPM8$_{LMNG-ca-icilin}$ from 69,436 particles yielded an EM map with a resolution of 2.98 Å.

For MmTRPM8$_{LMNG-Ca-icilin-PIP2}$, 3082 micrographs were collected and 1,081,783 particles were auto-picked and extracted with a binning factor of 3 before 2D classification. A total of 1,032,605 particles were selected for 2 rounds of 3D classification using the MmTRPM8$_{LMNG-ligand-free}$ map as the reference. Particles from two 3D classes were selected, combined, and reextracted to the pixel size of 1.014 Å, followed by 3D refinement with C4 symmetry and particle polishing via RELION 3.0. The final 3D reconstruction of MmTRPM8$_{LMNG-Ca-icilin-PIP2}$ from 57,439 particles yielded an EM map with a resolution of 3.21 Å.

For MmTRPM8$_{nanodisc-Ca-icilin}$, 1600 micrographs were collected and 830,781 particles were auto-picked and extracted with a binning factor of 3 before 2D classification. A total of 533,327 particles were selected for 2 rounds of 3D classification using the MmTRPM8$_{LMNG-ligand-free}$ map as the reference. Particles from three 3D classes were selected, combined, and reextracted to the pixel size of 1.014 Å, followed by 3D refinement with C4 symmetry and particle polishing via RELION 3.0. The final 3D reconstruction of MmTRPM8$_{nanodisc-Ca-icilin}$ from 131,232 particles yielded an EM map with a resolution of 2.52 Å.

For MmTRPM8$_{nanodisc-Ca-icilin-PIP2}$, 1602 micrographs were collected and 685,406 particles were auto-picked and extracted with a binning factor of 3 before 2D classification. A total of 412,376 particles were selected for 2 rounds of 3D classification using the MmTRPM8$_{LMNG-ligand-free}$ map as the reference. Particles from two 3D classes were selected, combined, and reextracted to the pixel size of 1.014 Å, followed by 3D refinement with C4 symmetry and particle polishing via RELION 3.0. The final 3D reconstruction of MmTRPM8$_{nanodisc-Ca-icilin-PIP2}$ from 62,791 particles yielded an EM map with a resolution of 3.04 Å.

The resolution was estimated by applying a soft mask around the protein density and the gold-standard Fourier shell correlation (FSC) = 0.143 criterion. Local resolution maps were calculated with RELION 3.0.

**Model building, refinement, and validation.** De novo atomic models were built based on the 2.88 Å resolution MmTRPM8$_{LMNG-Ca}$ density map in Coot[34]. The amino acid assignment was achieved on the basis of the clearly defined density for bulky residues (Phe, Trp, Tyr, and Arg) and the model of PmTRPM8$_{Ca}$ (PDB: 6O77) was used as a reference. Models were refined against cryo-EM maps using real-space refinement in PHENIX[35], with secondary structure and non-crystallography symmetry restraints applied. The initial cryo-EM density map allowed us to build an MmTRPM8$_{LMNG-Ca}$ model covering about 85% of the entire sequence. The models of MmTRPM8$_{LMNG-ligand-free}$, MmTRPM8$_{LMNG-Ca-icilin}$, MmTRPM8$_{LMNG-Ca-icilin-PIP2}$, MmTRPM8$_{nanodisc-Ca-icilin}$ and MmTRPM8$_{nanodisc-Ca-icilin-PIP2}$ were built using the

model of MmTRPM8$_{LMNG-Ca}$ as a template. The geometry statistics for models were generated using MolProbity[36]. All figures were prepared in PyMoL[37] or Chimera[38].

**Electrophysiology.** Patch-clamp recordings were performed with a HEKA EPC10 amplifier controlled by PatchMaster software (HEKA) in the whole-cell configuration. The membrane potential was held at 0 mV and the currents were elicited by two steps, 300 ms to +80 mV and followed by 300 ms to −80 mV. For whole-cell recording, serial resistance was compensated by 60%. The current was sampled at 10 kHz and filtered at 2.9 kHz. Patch pipettes were prepared from borosilicate glass and fire-polished to resistance of ~4 MΩ. Whole-cell patch-clamp measurements were performed 24–36 h after transfection at room temperature. A solution with 130 mM NaCl, 10 mM glucose, 0.5 mM CaCl$_2$ and 3 mM HEPES, pH 7.2 was used in both bath and pipette for whole-cell recordings.

The pipette with a whole-cell patch was placed in front of the perfusion tube outlet to ensure adequate perfusion. Ligands were perfused to membrane patch by a gravity-driven system (RSC-200, Bio-Logic). Bath and ligand solutions were delivered through separate tubes to minimize the mixing of solutions. Patch pipette with a membrane patch was placed directly in front of the perfusion tube outlet. Each membrane patch was recorded only once. All data points are mean ± s.e.m. ($n = 3–5$).

**All-atom molecular dynamic simulation.** The tetramer of the transmembrane domain of MmTRPM8$_{LMNG-Ca}$ or MmTRPM8$_{LMNG-Ca-icilin}$ (residues 722–1030) is embedded into the POPC lipid bilayer by using the CHARMM-GUI software packages[39]. The system is then solved in water. A periodic rectangular box with approximate dimensions of 140 × 140 × 110 Å was applied, which contains ~200,000 atoms. The parameters of protein, lipid, ligand, and ions are taken from CHARM36m force field[40]. The TIP3P model is chosen for water molecules[41]. 150 mM Na$^+$ and Cl$^-$ were added to neutralize the system. The energy of the system was minimized with protein position restraints of the backbone (4000 kJ/mol/nm$^2$) and side chains (2000 kJ/mol/nm$^2$), as well as lipid position and dihedral restraints (1000 kJ/mol/nm$^2$) using 5000 steps of the steepest descent. The simulation system was then pre-equilibrated using multi-step isothermal-isovolumetric (NVT) and isothermal-isobaric (NPT) conditions while decreasing the restraints at each step. Production simulations without restraints were generated with 2 fs time integration steps. The system temperature and pressure are controlled using the Nose-Hoover thermostat and Parrinello-Rahman barostat respectively. The LINCS algorithm is adopted to constrain the bond vibrations involving hydrogen atoms. Four independent simulation trajectories are carried out using the GROMACS 2021.4 package[42]. Analyses were performed using GROMACS 2021.4 package and the visual molecular dynamics (VMD) program[43].

**Reporting summary.** Further information on research design is available in the Nature Research Reporting Summary linked to this article.

## Data availability

The data that support this study are available from the corresponding authors upon reasonable request. The cryo-EM density maps have been deposited in the Electron Microscopy Data Bank (EMDB) under accession numbers EMD-32720 (MmTRPM8$_{LMNG-ligand-free}$), EMD-32721 (MmTRPM8$_{LMNG-Ca}$), EMD-32723 (MmTRPM8$_{LMNG-Ca-icilin}$), EMD-32722 (MmTRPM8$_{LMNG-Ca-icilin-PIP2}$), EMD-32724 (MmTRPM8$_{nanodisc-Ca-icilin}$), and EMD-32725 (MmTRPM8$_{nanodisc-Ca-icilin-PIP2}$). The coordinates have been in the RCSB Protein Data Bank (PDB) under accession codes 7WRA (MmTRPM8$_{LMNG-ligand-free}$), 7WRB (MmTRPM8$_{LMNG-Ca}$), 7WRD (MmTRPM8$_{LMNG-Ca-icilin}$), 7WRC (MmTRPM8$_{LMNG-Ca-icilin-PIP2}$), 7WRE (MmTRPM8$_{nanodisc-Ca-icilin}$), and 7WRF (MmTRPM8$_{nanodisc-Ca-icilin-PIP2}$). Source data are provided with this paper.

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

## Acknowledgements
Single particle cryo-EM data were collected at Center of Cryo-Electron Microscopy at Zhejiang University and the Cryo-Electron Microscopy Facility of Hubei University. We thank Dr. Xing Zhang and Dr. Shenghai Chang for support in facility access and data acquisition. This work was supported in part by the Ministry of Science and Technology (2020YFA0908501 and 2018YFA0508100 to J.G.), the National Natural Science Foundation of China (31870724 to J.G., 32122040, 31741067 and 31800990 to F.Yang.), Zhejiang Provincial Natural Science Foundation (LR19C050002 to J.G., LR20C050002 to F.Yang.), and the Fundamental Research Funds for the Central Universities (2021FZZX001-28 to J.G.). J.G. and F.Yang. are supported by MOE Frontier Science Center for Brain Science & Brain-Machine Integration, Zhejiang University. This work was supported by Alibaba Cloud.

## Author contributions
J.G., Y.X., and F.Yang. conceived and supervised the project. C.Z. and Y.X. did sample preparation, data acquisition, and structure determination. C.Z., Y.X., X.X., and W.Y. performed structure data analysis. L.X. performed electrophysiological studies. F.Ye and F.Yang performed molecular dynamic simulations. All authors participated in the data analysis and manuscript preparation.

## Competing interests
The authors declare no competing interests.
