## [Peer Review File · Nature Communications]

Structures of a Mammalian TRPM8 in Closed StateReviewers' Comments:

Reviewer #1:

Remarks to the Author:

The authors used cryo-electron microscopy to solve high-resolution structures of mouse TRPM8, a physiologically important transient receptor potential channel of the melastatin subfamily which, as a cold sensor, plays inter alia a fundamental role in thermal sensitivity. Previously, the Julius/Cheng and Lee labs had determined near-atomic structures of TRPM8 from birds which share a high degree of homology with the mouse orthologue used in this study. Correspondingly, the first structures of the mammalian TRPM8 (MmTRPM8) presented here in the presence of calcium and icilin, a well-known small-molecule agonist of this channel, and also in the absence of ligands are very much consistent with the previously published bird TRPM8 structures. The authors validate the icilin binding site using mutagenesis and electrophysiological recordings. The authors conclude that the location of both icilin and calcium binding sites in MmTRPM8 is essentially the same as in the bird orthologues.

The manuscript is well written and the figures are of high quality. The biochemical and cryo-EM approaches are standard and raise no questions.

The cryo-EM density maps have an overall good quality. Densities for icilin and calcium are convincing.

The abstract and introduction are somewhat misleading because no PIP2 density/binding was observed in any of the maps presented in this work. Although 'in the presence of [...] PIP2' is not entirely wrong, it creates an impression that PIP2 binding was in fact captured in this work. I strongly suggest to alter the text of the abstract and introduction to make this point clear.

In view of the previous work on bird orthologues and the proposed calcium coordination sites there, as well as on the relatively pronounced densities in cryo-EM maps in this work that overlap with the previously proposed calcium coordination sites, the binding of calcium ions in this work seems likely to be correct. Nonetheless, at the resolution of presented structures and without any supporting functional or computational data, it remains somewhat speculative to propose that small blobs of density necessarily represent calcium ions. The authors should consider to add more functional work, such as electrophysiological recordings, and MD simulations to strengthen their hypothesis about the location of calcium coordinating sites in order to not just rely on the previously published data for the bird orthologues. In the absence of additional data, the presence and location of the putative calcium ions needs to be evaluated and mentioned with more caution, emphasizing that the proposed locations are foremost based on observations made in previous work.

The density for icilin is convincing and is also supported by functional data.

It is not clear why PIP2 was added during nanodisc reconstitution since it is perfectly soluble in water. Please explain. Have the authors tried to add PIP2 at other steps during the preparation?

What is % homology between mentioned TRPM8 orthologues? It would be useful to provide a sequence comparison with the secondary structure elements and key residues highlighted. To my knowledge, the previously published *Parus major* TRPM8 shares ~82% sequence identity and >90% sequence similarity with the *Mus musculus* orthologue.

In Figure 1a, different resolution ranges/color-coding has been used to display the local resolution of maps. It would be more insightful for the reader to use the same scale for all maps to allow better comparison.

Lines 68, 69, 72, and 73: Please provide PDB IDs for the structures mentioned.

Line 516: 'd' in figure caption text should also be in bold to adhere with used style.

Table 1: Typo – capital 'C' in the last 'closed' or convert the rest to small letters.

Lines 332- 336: In view of a proper control, I do not understand why for the supposedly 'Ca²⁺-bound' sample, no exogenous Ca²⁺ was added (relying on the endogenously coordinated calcium or traces of calcium in the buffer), whereas a substantial amount of calcium (0.5 mM CaCl₂) was added to the Ca²⁺-icilin-bound sample? How can authors be confident that there is a residual calcium?

Reviewer #2:

Remarks to the Author:

This manuscript describes structures of mouse TRPM8 in several different states at resolutions ranging from 2.5 to 3.2 Å. It also describes electrophysiological recordings from WT and mutant forms of the channel to test the roles of specific residues observed in the structures to functional properties.

This study is of great interest because of the importance of TRPM8 in cold sensing, and because they lead to strikingly different conclusions about structural comparisons among TRPM family members than those suggested by previously published structures of bird TRPM8. The structural work is of high quality, and the interpretation overall is reasonable. The current recordings serve to back up the observations in the structures. The writing is generally quite good and clear.

There are only a few very minor changes in wording I would suggest be considered:

1. Page 8, the use of the word, "vacuuming" seems inappropriate. The authors seem to mean, "making space for." You cannot "vacuum" space.
2. Page 9, reference to Supplementary Fig. 10e. Menthol activation is used as a control, and it verifies that the mutant channel still has activity. However, the current amplitude seems to be considerably lower. Is this just random fluctuation of numbers of channels in different cells, or do the mutants have lower peak currents or lower menthol sensitivity?
3. Page 10, wording, "...difficult to be crossed," is awkward. Do the authors mean, "...from which it is difficult to transition?"
4. Page 11, wording, "Previously, structure 269 comparisons of PmTRPM8 ligand-free and PmTRPM8 Ca contribute the conformational difference...
Do the authors mean "attribute" rather than "contribute?"
5. Page 11, "Therefore, it is likely 275 that PmTRPM8Ca is also in the closed state (Supplementary Fig. 9h), not the desensitized state..." I would caution against interpreting previous structural results in terms of a state which is well-defined kinetically, but poorly defined structurally. Whether a state of an ion channel with a very narrow pore is "closed" or "desensitized" is nearly impossible to determine from a static structure.

Reviewer #3:

Remarks to the Author:

Zhao et. al. report structures of mouse TRPM8 in apo and agonist bound states. These new structures possibly correctly define the pose of icilin and clarify conformational states of reported bird TRPM8 structure. Overall, this study fall short to provide significant new insights into TRPM8 gating or pharmacology.

Structural determination and electrophysiology are solid. Molecular dynamic simulation could further confirm the experimental pose of icilin.

Reviewer #1 (Remarks to the Author):

The authors used cryo-electron microscopy to solve high-resolution structures of mouse TRPM8, a physiologically important transient receptor potential channel of the melastatin subfamily which, as a cold sensor, plays inter alia a fundamental role in thermal sensitivity. Previously, the Julius/Cheng and Lee labs had determined near-atomic structures of TRPM8 from birds which share a high degree of homology with the mouse orthologue used in this study. Correspondingly, the first structures of the mammalian TRPM8 (MmTRPM8) presented here in the presence of calcium and icilin, a well-known small-molecule agonist of this channel, and also in the absence of ligands are very much consistent with the previously published bird TRPM8 structures. The authors validate the icilin binding site using mutagenesis and electrophysiological recordings. The authors conclude that the location of both icilin and calcium binding sites in MmTRPM8 is essentially the same as in the bird orthologues.

We thank the reviewer's positive comments and constructive suggestions. We have collectively addressed all the reviewer's concerns. The following are our point-to-point responses to the reviewer's comments.

1. The abstract and introduction are somewhat misleading because no PIP₂ density/binding was observed in any of the maps presented in this work. Although 'in the presence of [...] PIP₂' is not entirely wrong, it creates an impression that PIP₂ binding was in fact captured in this work. I strongly suggest to alter the text of the abstract and introduction to make this point clear.

We thank the reviewer for pointing out the potential misleading of our wording. To avoid this misleading, we have removed the PIP₂ in both the abstract and introduction.

2. In view of the previous work on bird orthologues and the proposed calcium coordination sites there, as well as on the relatively pronounced densities in cryo-EM maps in this work that overlap with the previously proposed calcium coordination sites, the binding of calcium ions in this work seems likely to be correct. Nonetheless, at the resolution of presented structures and without any supporting functional or computational data, it remains somewhat speculative to propose that small blobs of density necessarily represent calcium ions. The authors should consider to add more functional work, such as electrophysiological recordings, and MD simulations to strengthen their hypothesis about the location of calcium coordinating sites in order to not just rely on the previously published data for the bird orthologues. In the absence of additional data, the presence and location of the putative calcium ions needs to be evaluated and mentioned with more caution, emphasizing that the proposed locations are foremost based on observations made in previous work.

We thank the reviewer's constructive suggestions. In the revision, we have performed all-atom Molecular dynamic (MD) simulations, which confirm that the Ca²⁺ binds at

this site stably throughout 250 ns simulations (Fig. 5c-f). In addition, we have tested the function of the Ca²⁺-coordinating residues in the Ca²⁺ potentiation on the icilin activation of MmTRPM8 using electrophysiology assays, which show that mutations on the Ca²⁺ coordination residues significantly reduced the potentiation effect of Ca²⁺ on MmTRPM8 with the presence of icilin (Fig. 5h). These computational and electrophysiological assays collectively support the assignment of Ca²⁺ in MmTRPM8. We have added the above analyses on Line 190-204.

3. It is not clear why PIP₂ was added during nanodisc reconstitution since it is perfectly soluble in water. Please explain. Have the authors tried to add PIP₂ at other steps during the preparation?

We thank the reviewer's constructive suggestion. In the nanodisc sample, the ion channel is embedded in the lipid molecules, which are further wrapped by the scaffold protein MSP1. It is believed that in the nanodisc the lipid molecules are less free than the detergent molecules in the detergent micelle. Therefore, to ensure the hydrophobic tails of PIP₂ to insert into the nanodisc, we added PIP₂ during the nanodisc reconstitution of TRPM8, even though the PIP₂ we used was water-soluble. In fact, this method has been successfully used in structure determinations of other ion channels with PIP₂ observed in the lipid nanodisc, such as KCNQ1-CaM-KCNE3 (Cell 2020, 180, 340–347) and KCNQ4-CaM (Neuron 2022, 110, 237–247).

For the TRPM8 in lipid nanodisc, we did not try to add PIP₂ at other steps. For the TRPM8 in the LMNG detergent, we directly added the water-soluble PIP₂ into the protein sample and incubated it at room temperature for 30 min before grid preparation. As TRPM8 structures determined in the presence of Ca+icilin+PIP₂ in the lipid nanodisc sample and LMNG detergent sample gave the same results, we believe the structure data are reliable.

4. What is % homology between mentioned TRPM8 orthologues? It would be useful to provide a sequence comparison with the secondary structure elements and key residues highlighted. To my knowledge, the previously published Parus major TRPM8 shares ~82% sequence identity and >90% sequence similarity with the Mus musculus orthologue.

We thank the reviewer's reminder. Both *Ficedula albicollis* TRPM8 (FaTRPM8) and *Parus major* TRPM8 (PmTRPM8) share 82% sequence identity and 91% sequence similarity to mouse TRPM8 (MmTRPM8). We have added the sequence alignment in Supplementary Fig. 1 and the above description on Line 69 in the second paragraph of the introduction section.

5. In Figure 1a, different resolution ranges/color-coding has been used to display the local resolution of maps. It would be more insightful for the reader to use the same scale for all maps to allow better comparison.

We thank the reviewer's suggestion. We have updated the color scale and used the same scale from 2.5 Å to 5.5 Å for all six maps.

6. Lines 68, 69, 72, and 73: Please provide PDB IDs for the structures mentioned.

We thank the reviewer's suggestion. We have added the PDB ID in the revised manuscript on Line 75, 79, and 80.

7. Line 516: 'd' in figure caption text should also be in bold to adhere with used style. We thank the reviewer's critical reading. The 'd' has been corrected.

8. Table 1: Typo – capital 'C' in the last 'closed' or convert the rest to small letters. We thank the reviewer's critical reading. This 'C' has been corrected.

9. Lines 332- 336: In view of a proper control, I do not understand why for the supposedly 'Ca²⁺-bound' sample, no exogenous Ca²⁺ was added (relying on the endogenously coordinated calcium or traces of calcium in the buffer), whereas a substantial amount of calcium (0.5 mM CaCl₂) was added to the Ca²⁺-icilin-bound sample? How can authors be confident that there is a residual calcium?

We thank the reviewer's advice. In the original manuscript, we assigned the Ca²⁺ ion in the structure of TRPM8_{LMNG-Ca} based on two facts.

(1) In the map of TRPM8_{ligand-free}, which was determined with the presence of 2 mM EGTA, a chelating agent showing a high affinity for Ca²⁺ ions, the density corresponding to the potential Ca²⁺ in the VSLD disappears. The comparison of TRPM8_{ligand-free} and TRPM8_{LMNG-Ca} at this potential Ca²⁺ site supports the attribution of this density to Ca²⁺.

(2) This Ca²⁺ site in the VSLD is conserved a subgroup of TRPM channels consisting of TRPM2, TRPM4, and TRPM5, as well as the previously reported structures of TRPM8 from *Ficedula albicollis* TRPM8 (FaTRPM8) and *Parus major* TRPM8 (PmTRPM8). Therefore, we are confident to assign the density as Ca²⁺.

We have listed these two lines of evidence in the original manuscript. As suggested by the reviewer, in the revision, we have provided two additional validations of this Ca²⁺ binding site in MmTRPM8.

(3) All-atom molecular dynamic (MD) simulations show that the Ca²⁺ binds at this site stably throughout 250 ns simulations (Fig. 5c-f).

(4) Electrophysiology assays show that mutations on the Ca²⁺-coordinating residues significantly reduced the potentiation effect of Ca²⁺ on MmTRPM8 with the presence of icilin (Fig. 5h).

Taken together, although we did not add Ca²⁺ in the sample, this isolated density in the VSLD of TRPM8_{LMNG-Ca} is attributed to Ca²⁺, either from endogenous Ca²⁺ or a trace amount of Ca²⁺ in solution. For the structure determination of TRPM8_{LMNG-Ca-icilin}, as the icilin resides near the Ca²⁺ which may potentially affect Ca²⁺ binding, to ensure there was sufficient Ca²⁺ in the protein sample, we added 0.5 mM CaCl₂.

Reviewer #2 (Remarks to the Author):

This manuscript describes structures of mouse TRPM8 in several different states at resolutions ranging from 2.5 to 3.2 Å. It also describes electrophysiological recordings from WT and mutant forms of the channel to test the roles of specific residues observed in the structures to functional properties.

This study is of great interest because of the importance of TRPM8 in cold sensing, and because they lead to strikingly different conclusions about structural comparisons among TRPM family members than those suggested by previously published structures of bird TRPM8. The structural work is of high quality, and the interpretation overall is reasonable. The current recordings serve to back up the observations in the structures. The writing is generally quite good and clear.

We thank the reviewer's insightful comments and positive evaluation. We have collectively addressed all the reviewer's concerns. The following are our point-to-point responses to the reviewer's comments.

There are only a few very minor changes in wording I would suggest be considered:

1. Page 8, the use of the word, "vacuuming" seems inappropriate. The authors seem to mean, "making space for." You cannot "vacuum" space.

We thank the reviewer's suggestion. "vacuuming" has been changed to "making" in the revision.

2. Page 9, reference to Supplementary Fig. 10e. Menthol activation is used as a control, and it verifies that the mutant channel still has activity. However, the current amplitude seems to be considerably lower. Is this just random fluctuation of numbers of channels in different cells, or do the mutants have lower peak currents or lower menthol sensitivity?

We thank the reviewer's comments. Compared with the WT MmTRPM8, these mutants have lower peak currents evoked by menthol and reduced menthol sensitivity. Nevertheless, the key information conveyed in these mutants was the icilin activation was virtually abolished in D802N, F839R and R842K, while Y1005F showed similar icilin activation as WT. Such a pattern of icilin response in these mutants can be adequately explained by the binding configuration of this molecule as observed in our cryo-EM structures.

3. Page 10, wording, "...difficult to be crossed," is awkward. Do the authors mean, "...from which it is difficult to transition?"

We thank the reviewer's comments. Yes, the reviewer's understanding is correct. To make this clearer, we have revised this sentence as "which is easy to be achieved but difficult to be crossed **over** *in vitro*"

4. Page 11, wording, "Previously, structure comparisons of PmTRPM8 ligand-free and PmTRPM8 Ca contribute the conformational difference...Do the authors mean "attribute" rather than "contribute?"

We thank the reviewer's correction. "contribute" has been replaced with "attributed" accordingly.

5. Page 11, "Therefore, it is likely that PmTRPM8Ca is also in the closed state (Supplementary Fig. 9h), not the desensitized state..." I would caution against interpreting previous structural results in terms of a state which is well-defined kinetically, but poorly defined structurally. Whether a state of an ion channel with a very narrow pore is "closed" or "desensitized" is nearly impossible to determine from a static structure.

We thank the reviewer's caution for the interpretation of the structural state of PmTRPM8Ca. To avoid overinterpretation, we have removed the "not the desensitized state" and reorganized this sentence as "Therefore, it is likely that PmTRPM8Ca is also in the closed state (Supplementary Fig. 11h), similar to the Ca²⁺-bound MmTRPM8_{LMNG-Ca} structure."

Reviewer #3 (Remarks to the Author):

Zhao et. al. report structures of mouse TRPM8 in apo and agonist bound states. These new structures possibly correctly define the pose of icilin and clarify conformational states of reported bird TRPM8 structure. Overall, this study fall short to provide significant new insights into TRPM8 gating or pharmacology.

We thank the reviewer's insightful comments and positive evaluation. We have addressed the reviewer's concerns. The following are our point-to-point responses to the reviewer's comments.

Structural determination and electrophysiology are solid. Molecular dynamic simulation could further confirm the experimental pose of icilin.

To confirm the pose of icilin modeled in the structure of MmTRPM8^{LMNG}-Ca-icilin, we performed all-atom MD simulations using the structure of MmTRPM8^{LMNG}-Ca-icilin as an initial model. Throughout the 250 ns simulation, the icilin binds stably at its original site, with an RMSD within 4 Å (Fig. 6e), and a distance of 3–7 Å between the N21 atom of icilin and the CG atom of F839 (Fig. 6f). Meanwhile, the Ca²⁺ also binds stably at its original site, with an RMSD within 3 Å (Fig. 6g), and a distance of 3–4 Å between the Ca²⁺ and the CG atom of D802 in most time of the simulations (Fig. 6h). These computational data provide additional support for our assignment of icilin based on the map density.

We have added the above data in Fig. 6e-h and text on Line 222-229.

Reviewers' Comments:

Reviewer #1:

Remarks to the Author:

I thank the authors for their thorough review of their manuscript. The authors have addressed all of my comments and I have no other comments.

Reviewer #2:

Remarks to the Author:

The revisions in the current version successfully address the concerns raised in the previous round of review.